# Circular Economy Approach in Phosphorus Fertilization Based on Vivianite must be Tailored to Soil Properties

Tolulope Ayeyemi<sup>1</sup>, Ramiro Recena<sup>2</sup>, Ana María García-López<sup>1</sup>, José Manuel Quintero<sup>1</sup>, María Carmen del Campillo<sup>3</sup>, and Antonio Delgado<sup>1</sup>

- <sup>1</sup>Department of Agronomy, ETSIA, University of Seville, Seville, 41013, Spain
  - <sup>2</sup>Department of Aerospace Engineering and Fluid Mechanics, University of Seville, Seville, 41013, Spain.
  - <sup>3</sup>Department of Agronomy, ETSIAM, University of Córdoba, Campus de Rabanales, 14071 Córdoba, Spain

Correspondence to: Tolulope Ayeyemi (toluopeayeyemi@gmail.com)

Abstract. Although there is relevant knowledge based on the effect of soil properties on the efficiency of common commercial fertilizers, this effect remains poorly understood for the use of vivianite from water purification as an innovative P fertilizer meeting a circular economy approach. This study aimed to evaluate the effect of soil properties on the efficiency of vivianite recovered from water purification as a P fertilizer and to provide practical recommendations for its effective use. Vivianite and a soluble mineral P fertilizer (superphosphate) were compared at two P application rates (50 and 100 mg P kg<sup>-1</sup>) in soils ranging widely in properties in a pot experiment using wheat. Soluble P fertilizer provided the best results in terms of dry matter (DM) yield, P uptake, and Olsen P in soils, while vivianite led to the best results of DTPA extractable Fe in soils after crop harvest. The application of vivianite as a P fertilizer was more efficient in acidic soils (pH  $\leq$  6.6). The effect of vivianite on dry matter (DM) yield was equivalent on average to 26 or 40 %, depending on the rate, of the same amount of soluble fertilizer in these acidic soils (i.e., P fertilizer replacement value -PFRV- on DM basis), it being around 50 % in some cases. The effect on Olsen P in soil was equivalent, on average, to 49 or 61 %, depending on the rate, of the same amount applied as soluble mineral fertilizer in acidic soils. This can be explained by the increased solubility of this fertilizer product under acidic conditions, supported by the highest increase in DTPA extractable Fe in these soils. Acidic soils were those with initial Olsen P below the threshold value for fertilizer response (TV). However, PFRV on different approaches (DM, P uptake, and Olsen P) decreased more consistently with increased values of the difference between initial Olsen P and TV (46 to 87 % of the variance explained) than with increased pH. This reveals that, besides soil pH, a low P availability to plants can trigger plant and microbial mobilization mechanisms, leading to increased efficiency of vivianite as a P fertilizer. These results are promising for the use of vivianite from wastewater treatment as a P fertilizer, the application of which should be adapted to the soil properties, and is especially recommended for acidic P-deficient soils.

# 1 Introduction

Phosphorus (P) is an essential nutrient required for optimal crop production (Balemi and Negisho, 2012; Sharma et al., 2013; Recena et al., 2017). Phosphorus fertilizers are majorly derived from phosphate rock (PR), which is a non-renewable and

strategic resource (Recena et al., 2022; Ayeyemi et al., 2023; 2024). Currently, 82% of PR is used for manufacturing phosphate fertilizers (Schröder et al., 2011; Heckenmüller et al., 2014). Phosphate rock production is expected to peak in the current century (Cordell et al., 2009; Keyzer, 2010), posing a serious constraint to global food security. The recent discovery of huge PR deposits in Norway (The Economist, 10 June 2023; Hernández-Mora et al., 2024) would allow us to think about a change in the situation concerning the use of P resources. However, new industrial uses (e.g. production of batteries) would lead to an increase in consumption of this non-renewable resource, with expected constraints for agricultural production (García-López et al., 2025). Thus, ensuring food security for an increasing world population, with an estimated rise to nine billion by 2050 (United Nations, 2017), makes it crucial to explore alternative sources of P fertilizers aside from PR to support crop production.

40

The productivity of soils is dependent on their physical, chemical, and biological properties (Delgado and Gómez, 2016; Bibi et al., 2023), which are widely different globally due to soil-forming factors and processes (Mahdi and Uygur, 2018). These properties and their interactions govern the availability of nutrients in the soil rhizosphere (Jiang et al., 2009). Soil properties determine the reaction of applied fertilizers depending on their chemical form and, consequently, the response of crops to their application (Bindraban et al., 2015). Hence, crops grown on different soils respond differently to fertilizer application (Olaniyan et al., 2011). It is well-known that P adsorption and precipitation in soils are strongly dependent on their physical and chemical properties, such as texture, soil pH, iron (Fe) and aluminium (Al) oxides, organic matter, and carbonates (Pizzeghello et al., 2016; Zhang et al., 2019; Sun et al., 2022). Thus, these properties affecting P reactions in soil, together with the nature of fertilizers, will determine the use of applied P by crops (Delgado and Scalenghe, 2008). Organic P present in soil or applied as fertilizers can also be a direct source of available P through its mineralization and the turnover of microbial biomass (Recena et al., 2015; 2018; Bueis et al., 2019). Thus, the introduction of new and alternative P fertilizer products into cropping systems should consider the significant role that soil properties play in the reactions of P and, consequently, in the availability of native and applied P in soils. Regarding P fertilizer application, the definition of threshold values for a given soil P test, such as the Olsen P test, is necessary to predict crop yield response to fertilizer application (Recena et al., 2016; 2022). The threshold value is the limit of the soil P test above which soils are not responsive to P fertilizer application (Syers et al., 2008). This is sometimes referred to as the P-critical value. An efficient use of P from an agronomic and environmental standpoint can be achieved if the P threshold value is taken into account in P fertilizer strategies and management (Svers et al., 2008).

A sustainable strategy to manage P resources is through the recycling of P from all current waste streams throughout the whole food system, including production, processing and consumption (Cordell et al., 2009; Recena et al., 2022). Urban wastewater constitutes an important source of recoverable phosphorus, potentially contributing 15–20% towards meeting the global yearly phosphorus demand (Wu et al., 2019). Nowadays, in the European Union (EU), recovered P currently displaces only about 0.5% of the use of conventional phosphate fertilizers, although the maximum potential is estimated at up to 13% of total EU phosphate fertilizer imports (Muys et al., 2021; Recena et al., 2022; Soo and Shon, 2024). Therefore, it is highly relevant to consider the importance of P recovery from wastewater for phosphorus sustainability in the EU, in alignment with the 5R strategy proposed by Withers et al. (2015) for sustainable P management. Vivianite, a Fe<sup>2+</sup> phosphate mineral (Fe<sub>3</sub>(PO<sub>4</sub>)<sub>2</sub>

.8H<sub>2</sub>O), forms under reducing conditions in wastewater treatment facilities (Wilfert et al., 2018), aquatic sediments, drained agricultural areas (Egger et al., 2015; Dijkstra et al., 2016; Rothe et al., 2016), and in waterlogged soils (Heiberg et al., 2012; Nanzyo et al., 2013). Vivianite is now gaining attention as a potential P fertilizer (Yaya et al., 2015; Wu et al., 2019), although its phosphorus content, around 10%, depends on its crystallinity and the recovery methods employed. A recent study by Ayeyemi et al. (2023) revealed that industrially produced vivianite has a replacement value, i.e. an equivalence in dry matter production, to 50-75% of the same amount of superphosphate. Eshun et al. (2024) demonstrated that vivianite produced with the use of Fe-reductant microorganisms was an efficient P fertilizer. Fodoué et al. (2015) and Jowett et al. (2018) found that vivianite application enhanced the growth and yield of bean and maize plants, respectively, and showed that P recovered from wastewater in the form of vivianite was equivalent or even superior to conventional superphosphate fertilizers. Vivianite can offer several advantages over conventional phosphate fertilizers, primarily by addressing the relatively low P uptake and use efficiency in crops, which often leads to overuse of soluble phosphate fertilizers and the resulting environmental impact and waste of this non-renewable resource (Monreal et al., 2016). Besides, vivianite may have a more favorable impact on soil microbial activity. A recent study by Faller et al. (2025) indicates that vivianite can act as a sustainable phosphate fertilizer that preserves the microbial potential for P cycling, promoting microbial taxa associated with P availability without significantly altering soil microbial community composition. This contrasts with mineral phosphorus fertilizers, which tend to affect the soil microbial communities, including P mobilizing bacteria (Liu et al., 2024; Deinert et al., 2025). However, the extraction costs of vivianite can exceed those of conventional fertilizer production, posing a disadvantage to its use. Xie et al. (2023) demonstrated through a comprehensive life-cycle analysis that the social costs associated with vivianite application are equal to or even lower than those of conventional phosphate fertilizer use.

Despite its potential, limited information is available on vivianite as a phosphate fertilizer, and the influence of soil properties on its effectiveness remains largely unknown. Soil environmental conditions are dynamic and heterogeneous, exerting significant influence on the redox transformation of iron minerals, a process closely linked to phosphorus mobilization and immobilization (Strong et al., 2004; Or et al., 2007; Peth et al., 2008; Yang et al., 2022). In this study, we investigated the impact of soil properties on the effectiveness of vivianite obtained from water purification as a P fertilizer using a pot experiment. We aimed to (1) evaluate the efficiency of using vivianite compared to mineral fertilizer based on the P fertilizer replacement value (PFRV); (2) explore the major properties of soil affecting P dynamics of vivianite; and (3) identify how dissolution of vivianite affects P availability to plants. This will allow us to demonstrate the possible use of vivianite as a P fertilizer under different soil conditions, leading to proper recommendations of its usage depending on soil properties.

#### 2 Material and Methods

#### 95 **2.1 Soils**


Twelve soil samples were collected from the surface horizon of typical soils developed under the Mediterranean climate. According to the Soil Taxonomy (Soil Survey Staff, 2014), selected soils were classified into Inceptisols, Alfisols, Vertisols,

and Mollisols. This selection included calcareous and non-calcareous soils. In each selected location, a square of 10 m x 10 m was defined with homogeneous surface properties in terms of color, texture, and structure. Then, 10-12 subsamples of the surface layer (0–20 cm) were randomly taken at sampling points. To this end, in each sampling point (1 m<sup>2</sup>), eight soil cores (50 mm diameter) were taken to obtain a subsample, and after that, all the subsamples from each sampling point were mixed to obtain a composite sample. The soils were air-dried, clods and lumps broken, and thereafter passed through a 2 mm sieve for laboratory analysis and sieved to <4 mm for pot experiment to avoid excessive destruction of soil structure that may affect crop performance in pots. Soils were analyzed for particle size distribution according to Gee and Bauder, (1986), organic C by the oxidation method of Walkley and Black, (1934), total CaCO<sub>3</sub> equivalent (CCE) by the calcimeter method, pH, and electrical conductivity in water at a soil: extractant ratio of 1:2.5, and the cation exchange capacity (CEC) by using 1 M NH<sub>4</sub>OAc buffered at pH 7 (Sumner and Miller, 1996). Oxalate extraction was performed to release Fe in poorly crystalline Fe oxides (Fe<sub>ox</sub>), and citrate-bicarbonate-dithionite to release Fe in crystalline oxides (Fe<sub>d</sub>) according to Recena et al. (2015). Olsen P was used as a soil P test, i.e, an availability index to assess the response to P fertilizer. It was determined by weighing two grams of soil into 50 mL Falcon tubes, after which 40 mL of 0.5M NaHCO<sub>3</sub> at pH 8.5 was added. The mixture was shaken in a mechanical end-over-end shaker for 30 minutes at 180 rpm. Subsequently, the suspension was centrifuged for 10 min at 900 g. The P concentration of the extract was determined by the colorimetric method of Murphy and Riley (1962) using a spectrophotometer at 882 nm. The DTPA (Diethylenetriaminepentaacetic acid) extractable Fe determination was carried out according to Lindsay and Norvell (1978) with slight modifications as Fe availability index. To this end, five grams of soil were weighed into 50 mL Falcon tubes, and 20 mL of DTPA/CaCl<sub>2</sub> TEA (triethanolamine) was added. This suspension was stirred for 2 h at 160 rpm. The suspension was then placed in the centrifuge for 15 min at 900 g. The Fe concentration of the extract was determined by atomic absorption spectrometry. Since the soils have very different properties, it is expected that the Olsen P threshold value (TV), i.e., the value above which no response in yield is expected with P fertilization, ranges widely among soils (Recena et al., 2016). This TV was calculated according to the model proposed by Recena et al. (2022). The equation of the model is Y = 43.7 - 0.016 Clay -3.81 pH. To assess the available P status of soils, the Olsen P value was compared with the specific TV for each soil. The more negative this difference between current Olsen P in soil and TV (Olsen P – TV) is, the more deficient the soil is in P. The detailed properties of the soils used in this experiment are shown in Table 1.

# 2.2 Fertilizers








Two fertilizer products were studied in this experiment: (i) Water Purification Vivianite (WPV) obtained from Wetsus (European Centre of Excellence for Sustainable Water Technology) from Leeuwarden, the Netherlands, and (ii) Superphosphate as a reference P fertilizer: Ca(H<sub>2</sub>PO<sub>4</sub>)<sub>2</sub>.H<sub>2</sub>O.

The elemental composition of the WPV (Table 2) was determined by ICP-OES after acid digestion, except for C and N; these two elements were determined in an elemental analyzer. The Fe<sup>2+</sup> to Fe<sup>3+</sup> ratio was determined by Mossbauer spectroscopy and X-ray photoelectron spectroscopy (XPS). This ratio is relevant since Fe<sup>3+</sup> compounds are less soluble and contribute little to nutrient supply to crops (Ayeyemi et al., 2023).

# 2.3 Experimental Design




A pot experiment was conducted using durum wheat (*Triticum durum* L. cv. Amilcar) under controlled conditions in a growing chamber. The experiment was arranged in a Randomized Complete Block Design (RCBD) with three replicates. Each replicate corresponded to a pot with one wheat plant. Two factors were involved in the experiment: Soil type (12) and P fertilizer treatment, involving the two fertilizers described above, at two rates (50 and 100 mg P kg<sup>-1</sup>), and a non-fertilized control. The lowest P rate was selected since it is generally believed that plants respond to fertilizer application at this rate in P-poor growing media in pot experiments (García-López et al., 2016). The highest rate was chosen to check the impact of a high rate on P absorption and availability in the growing medium.

The growing media was prepared by mixing fertilizer products with 300 g of soil and placed in cylindrical polyethylene pots with a volume of 350 mL (height 150 mm; diameter 55 mm). The mixing of fertilizer products (in powder form) with soil was carried out three days before transplanting the wheat seedlings. Wheat seeds were pregerminated by sowing in a nursery for 15 days, after which they were transplanted into already prepared growing assays. The assay was placed in the growing chamber with temperatures of 25°C/16°C day/night and irrigation till 70% of the water holding capacity of the soils, with replenishment by weight loss. Within the first two days of transplanting the wheat seedlings, irrigation was conducted only with water, after which a P-free nutrient solution (Hoagland type) was applied on a regular basis. The composition of this

**Table 1. Soil Properties** 

| Soil    | Clay          | Silt          | Sand           | CCE           | ос            | pН            | EC              | CEC                       | Feox          | $Fe_{d}$      | Olsen<br>P     | TV*            | Olsen<br>P – TV | DTPA<br>Fe      |
|---------|---------------|---------------|----------------|---------------|---------------|---------------|-----------------|---------------------------|---------------|---------------|----------------|----------------|-----------------|-----------------|
|         | _             |               | g              | kg-1          |               | _             | $\mu S/cm^{-1}$ | cmolc<br>kg <sup>-1</sup> |               |               |                |                | mg k≨           | g <sup>-1</sup> |
|         |               |               |                |               |               | Cal           | careous so      | ils                       |               |               |                |                |                 |                 |
| Soil 1  | 362           | 140           | 180            | 330           | 6.2           | 8.30          | 284             | 34.3                      | 0.68          | 5.8           | 17.0           | 6.3            | 10.7            | 9.8             |
| Soil 2  | 245           | 126           | 488            | 140           | 8.8           | 8.50          | 134             | 17.0                      | 0.30          | 10.5          | 14.4           | 7.4            | 7.0             | 5.3             |
| Soil 3  | 228           | 164           | 463            | 139           | 8.4           | 8.70          | 201             | 13.0                      | 0.36          | 14.5          | 14.5           | 6.9            | 7.6             | 6.5             |
| Soil 4  | 168           | 158           | 480            | 184           | 7.3           | 8.80          | 157             | 9.7                       | 0.24          | 12.0          | 8.9            | 7.5            | 1.4             | 5.0             |
| Soil 5  | 222           |               |                | 212           | 9.7           | 8.10          | 278             | 16.3                      | 0.66          | 6.7           | 16.3           | 9.3            | 7.0             | 7.8             |
| Soil 6  | 120           | 150           | 630            | 43            | 12.0          | 8.26          | 173             | 28.5                      | 1.55          | 8.4           | 16.9           | 10.3           | 6,6             | 8.6             |
| Mean±SD | 224±81<br>.8  | 147±15<br>.06 | 448±16<br>4.16 | 175±95<br>.31 | 8.73±2<br>.01 | 8.44±0.<br>27 | 204±63.<br>17   | 19.8±9.<br>53             | 0.63±0.<br>49 | 9.65±3<br>.31 | 14.67±<br>3.05 | 7.95±1.<br>53  | 6.74±3.<br>36   | 7.17±1.9<br>0   |
|         |               |               |                |               |               | Non-          | calcareous      | soils                     |               |               |                |                |                 |                 |
| Soil 7  | 92            | 150           | 769            | 0             | 4.1           | 6.44          | 45              | 7.3                       | 1.47          | 6.7           | 16.4           | 17.7           | -1.3            | 83.0            |
| Soil 8  | 62            | 170           | 771            | 0             | 13.2          | 6.60          | 30              | 11.1                      | 0.90          | 13.5          | 8.4            | 17.6           | -9.2            | 36.7            |
| Soil 9  | 155           | 212           | 632            | 0             | 8.8           | 5.84          | 40              | 12.1                      | 1.74          | 17.4          | 7.3            | 19.0           | -11.7           | 47.7            |
| Soil 10 | 130           | 180           | 690            | 0             | 5.8           | 5.76          | 84              | 10.8                      | 1.40          | 13.0          | 12.0           | 19.7           | -7.7            | 44.5            |
| Soil 11 | 388           | 156           | 443            | 0             | 15.6          | 7.86          | 138             | 58.6                      | 2.59          | 17.5          | 20.7           | 7.5            | 13.2            | 17.8            |
| Soil 12 | 274           |               |                | 0             | 6.4           | 7.90          | 224             | 15.4                      | 0.74          | 17.1          | 13.8           | 9.2            | 4,6             | 4.8             |
| Mean±SD | 183±12<br>3.9 | 173±24<br>.47 | 661±13<br>5.06 | 0             | 8.98±4<br>.52 | 6.73±0.<br>94 | 93.5<br>±75.30  | 19.2±1<br>9.47            | 1.47±0.<br>66 | 14.2±4<br>.18 | 13.1±5.<br>02  | 15.12±<br>5.33 | 3.34±1<br>0.01  | 39.08±2<br>7.07 |

CCE. Ca carbonate equivalent; ACCE. active Ca carbonate equivalent; EC. electrical conductivity; OC, organic carbon, CEC. cation exchange capacity; Ca. Mg. K. and Na. exchangeable cations;  $Fe_{ox}$ . oxalate extractable Fe;  $Fe_{dx}$ . citrate-bicarbonate-ditthionite extractable Fe; DPTA Fe. \*TV. Threshold Value calculated as: Y = 43.7–0.016 Clay – 3.81 pH (Recena et al., 2022)

Table 2. Elemental composition of the vivianite used in the experiment and percentage of total Fe as  $Fe^{2+}$  and  $Fe^{3+}$  according to Mossbauer spectroscopy and X-ray photoelectron spectroscopy (XPS)

|    |    |     |      |     |     |     |      |      |      | Moss             | bauer | X         | PS               |
|----|----|-----|------|-----|-----|-----|------|------|------|------------------|-------|-----------|------------------|
| C  | N  | P   | K    | Ca  | Mg  | Fe  | Zn   | Mn   | Cu   | Fe <sup>2+</sup> | Fe³+  | $Fe^{2+}$ | Fe <sup>3+</sup> |
|    |    |     |      | g k | g-1 |     |      |      |      |                  |       |           |                  |
| nd | nd | 108 | 0.25 | 8.9 | 9.2 | 280 | 0.16 | 1.14 | 0.04 | 75               | 25    | 41        | 59               |

nd = not detectable; XPS. X-ray Photoelectron Spectroscopy

nutrient solution was (all concentrations in mmol L-1): MgSO<sub>4</sub> (2), Ca(NO<sub>3</sub>)<sub>2</sub> (5), KNO<sub>3</sub> (5), KCl (0.05), Fe- EDDHA (0.02), H<sub>3</sub>BO<sub>3</sub> (0.024), MnCl<sub>2</sub> (0.0023), CuSO<sub>4</sub> (0.0005), ZnSO<sub>4</sub> (0.006), and H<sub>2</sub>MoO<sub>4</sub> (0.0005). The wheat plants were harvested 58 days after transplanting at the ripening stage.

# 2.3 Collection of Soil and Plant Samples

At the end of the experiment, bulk soil samples (the entire soil samples in the pots) were collected for Olsen P and DTPA Fe analyses as described above. These samples were dried and milled to pass through a 2 mm screen. The roots and shoots of the wheat plants were also collected separately. Wheat root and shoot plant samples were placed in a forced-air oven dryer at 65°C for 72 h, after which the dry matter (DM) in each organ was determined.

# 2.3.1 Plant Samples Analysis





Root and shoot wheat samples were ground. Subsequently, wet acid digestion was carried out. To this end, 50 mg of plant materials were placed in glass test tubes, and 1 mL HNO<sub>3</sub> was added. The mixture was left to stand overnight. This was placed in an open block digest the next morning and heated to temperatures of 120°C -130°C until the plant materials were fully digested and clear. 10 mL of Milli-Q water was added and allowed to stand overnight, after which the P concentration in the digest was determined by ICP-OES. The total P uptake by plants was determined as the sum of the product of the dry weight of each organ and its P concentration. The P fertilizer replacement value (PFRV) of vivianite was adapted from Hijbeek et al. (2018) as the amount of commercial mineral P fertilizer (superphosphate) saved or replaced when using an alternative fertilizer (in this case, vivianite) while attaining the same yield, P uptake, or Olsen P in soils. This gives an idea of equivalence if expressed on a percentage basis. It is expressed as the kg of commercial mineral fertilizer that provides the same effect as 100 kg of alternative fertilizer. Thus, it can be interpreted as the percentage of commercial mineral fertilizers that can be replaced by alternative fertilizers. It was estimated on a DM basis for each P rate following Eq. (1):

$$PFRV_{DM} = \frac{DM_v - DM_c}{DM_s - DM_c} \tag{1}$$

where  $DM_v$  is the DM yield with vivianite,  $DM_c$  is the average DM in the non-fertilized control, and  $DM_s$ , the average DM in the superphosphate treatment at the same P rate as vivianite.

The PFRV was estimated on a P uptake basis for each P rate following Eq. (2):

$$PFRV_{P \text{ Uptake}} = \frac{Puptake_v - Puptake_c}{Puptake_s - Puptake_c}$$
 (2)

Where Puptake<sub>v</sub> is the P uptake by crop with vivianite, Puptake<sub>c</sub> is the average P uptake in the non-fertilized control, and Puptake<sub>s</sub>, the average P uptake in the superphosphate treatment at the same P rate as vivianite.

The PFRV was also estimated on an Olsen P basis following Eq. (3):





$$PFRV_{Olsen P} = \frac{Olsen P_{v} - Olsen P_{c}}{Olsen P_{s} - Olsen P_{c}}$$
(3)

where Olsen  $P_v$  is the Olsen P with vivianite, Olsen  $P_c$  is the average Olsen P of the non-fertilized control and Olsen  $P_s$ , the average Olsen P in the superphosphate treatment at the same rate as vivianite

#### 185 **2.4 Statistical Analysis**

Statistical analysis was performed with Statgraphics Centurion 18 (Statgraphics Technologies, 2018). The effect of factors (P fertilizer treatment and soils as fixed factors) on DM yield and P uptake was assessed by means of a two-way analysis of variance (ANOVA). To assess the effect of soil on the different PFRV indices studied and the increase of DTPA extractable Fe, one-way ANOVA was performed for each P rate independently. Before ANOVA, normality and homogeneity of variance were assessed with the use of the Smirnov–Kolmogorov and Levene tests, respectively. Power transformations were performed when one or both tests were not passed. Mean separation was conducted using the Tukey Honest Significant Difference (HSD) test at P < 0.05. If the interaction between factors was significant, the effects of the main factors were not discussed since the effect of one factor depends on the level of the other. To assess the differences in PFRV indices and increase in DTPA extractable Fe between calcareous and non-calcareous soils, an ANOVA with the factor soil type (i.e. calcareous or non-calcareous) was performed and means compared according to the Tukey test as above. To assess the differences between soils with pH < 6.6 (n = 4) and those with pH > 7.86 (n = 8), the non-parametric Kruskal-Wallis test was performed. In this case, medians were compared according to the procedure of Bonferroni. Regression and correlation analysis were performed using the same software to see relationships between different soil properties.

# 3 Results

# 200 3.1 Soil Properties

There was wide variation in the properties of the set of soils used in this experiment (Table 1), especially clay content and calcium carbonate equivalent (CCE), which are relevant properties affecting P dynamics in soils. The 12 soils used were grouped into two broad categories: six calcareous and six non-calcareous. The pH of the non-calcareous soils varied from 5.76 to 7.90, while in the calcareous soils, it ranged from 8.10 to 8.80. The DTPA extractable Fe content of the calcareous soils ranged from 5.0 to 9.8 mg kg<sup>-1</sup>, while those of non-calcareous soils were higher, ranging from 17.8 to 83.0 mg kg<sup>-1</sup>. The organic

matter content of all soils varied from 4.1 to 12.0 g kg<sup>-1</sup>, while the clay content of all soils varied from 62 to 388 g kg<sup>-1</sup>. There was also a wide variation in the Olsen P value of the soils from 7.3 to 20.7 mg kg<sup>-1</sup>. The Olsen P – TV of the calcareous soils ranged from 1.4 to 10.7, while those of non-calcareous soils ranged from -11.7 to 13.2. The Olsen P – TV values were positively correlated with pH (r = 0.83; P < 0.001) and clay content (r = 0.77; P < 0.01). The four non-calcareous soils with pH lower than 6.6 were the soils with negative values of Olsen P – TV. The oxalate and dithionite extractable Fe were negatively correlated with carbonate content (r = -0.62 and -0.59, respectively; P < 0.05 in both cases).

#### 3.2 Effect of Soils and Fertilizer Treatments on Crops and Soil






The dry matter (DM) yield, total P uptake by plants, and Olsen P and DTPA extractable Fe in the soil after crop were significantly affected by the interaction between both factors, soil and fertilizer treatment (P < 0.001 in all cases; Table 3). This means that the effect of fertilizer treatments depends on soil. Overall, soluble mineral fertilizer (superphosphate) provided the best results in terms of DM, P uptake, and Olsen P; meanwhile, vivianite led to the best results of DTPA extractable Fe in soil after crop (Table 3). The vivianite treatments led to increased DM yield relative to non-fertilized control in soils 8, 4, 9 and 10 (Table S1). In these soils and in soil 3, vivianite treatments slightly increased P uptake when compared with the control (Table S1). In soils 8, 4, 7, 9 and 10, vivianite led to higher Olsen P after crop than control, and in soils 9 and 10, the highest rate of vivianite promoted higher Olsen P than the lowest rate of mineral soluble P fertilizer (Table S1). The increase of DTPA extractable Fe with vivianite relative to superphosphate and control was particularly evident in soils 8, 9, and 10 (Table S1).

With vivianite at 50 mg P kg<sup>-1</sup>, the effect of soil was not significant for the P fertilizer replacement value on a DM basis (PFRV<sub>DM</sub>). Soil had a significant effect on PFRV on a P uptake basis (PFRV<sub>P Uptake</sub>, P < 0.001) and on an Olsen P basis (PFRV<sub>Olsen P</sub>, P < 0.001) at this lowest vivianite rate. At the highest rate (100 mg P kg<sup>-1</sup>), PFRV<sub>DM</sub>, PFRV<sub>P Uptake</sub>, and PFRV<sub>Olsen P</sub> of vivianite were significantly affected by soil (P < 0.05, 0.01 and 0.001, respectively) (Table S2). In order to better understand the effect of soils on the different PFRV indices used, the relationships of these variables with soil properties were studied. These PFRV indices provide a relative comparison of efficiency with the mineral soluble P fertilizer.

#### 230 3.3 Phosphorus Fertilizer Replacement Value on a Dry Matter Basis

When soils were discriminated by carbonate content, differences in this index were not significant. Meanwhile, soils with pH lower than 6.6 showed PFRV<sub>DM</sub> at the highest P rate, significantly higher than those with pH >7.86 (Table 4). However, PFRV<sub>DM</sub> at both rates was not correlated with pH. It was negatively correlated with clay content (r = -0.79; P < 0.01) and the value of the difference between soil Olsen P and its threshold value (Olsen P – TV; r = -0.71; P < 0.05) at the lowest P fertilizer rate (50 mg P kg<sup>-1</sup>) (in both cases an outlier with PFRV<sub>DM</sub> > 540 was excluded), meanwhile it was negatively correlated only with Olsen P – TV (r = -0.68; P < 0.05) at the highest P fertilizer rate (100 mg P kg<sup>-1</sup>). Thus, Olsen P – TV was the only variable that was correlated with PFRV<sub>DM</sub> at both P fertilizer rates, and PFRV<sub>DM</sub> decreased significantly with increased values of the difference (Olsen P – TV; Figure 1). It was observed that for soils with Olsen P – TV less than 0 (which were also the soils with pH < 6.6), PFRV<sub>DM</sub> was always positive at both P fertilizer rates, with average values of 26 and 40 % at the lowest

and the highest P fertilizer rates, respectively. However, for soils with Olsen P – TV higher than 0, most of the soils showed negative PFRV<sub>DM</sub> at the lowest P rate and three soils at the highest rate (Figure 1). At the lowest P fertilizer rate, the soil with an Olsen P – TV of - 9.2 resulted in a PFRV<sub>DM</sub> of 54%. At the highest rate of P fertilizer application, soils with Olsen P – TV < - 5 showed PFRV<sub>DM</sub> of around 50 % (Figure 1).

Table 3. Effect of fertilizer treatments and soil on dry matter yield (DM) and P uptake by crop. and Olsen P and DTPA extractable Fe after crop

| Factor                   | n  | DI    |                   | P up | take |                   | Ols  | en P |       | DTPA Fe            |         |       |     |  |
|--------------------------|----|-------|-------------------|------|------|-------------------|------|------|-------|--------------------|---------|-------|-----|--|
|                          |    | g pla | int <sup>-1</sup> |      | mg p | lant <sup>-</sup> | 1    |      | — mş  | g kg <sup>-l</sup> |         |       |     |  |
| Fertilizer treatment     |    |       |                   |      |      |                   |      |      |       |                    |         |       |     |  |
| Control                  | 36 | 1.21  | $\pm$             | 0.07 | 1.64 | $\pm$             | 0.15 | 14.7 | $\pm$ | 1.3                | 14.7    | $\pm$ | 1.3 |  |
| Superphosphate 100       | 36 | 1.74  | $\pm$             | 0.08 | 4.67 | $\pm$             | 0.27 | 48.1 | $\pm$ | 3.0                | 48.1    | $\pm$ | 3.0 |  |
| Superphosphate 50        | 36 | 1.67  | $\pm$             | 0.07 | 3.46 | $\pm$             | 0.24 | 28.3 | $\pm$ | 1.8                | 28.3    | $\pm$ | 1.3 |  |
| Vivianite 100            | 36 | 1.28  | $\pm$             | 0.06 | 1.63 | $\pm$             | 0.10 | 17.7 | $\pm$ | 1.1                | 17.7    | ±     | 1.  |  |
| Vivianite 50             | 36 | 1.27  | ±                 | 0.06 | 1.67 | ±                 | 0.12 | 15.8 | ±     | 1.0                | 15.8    | ±     | 1.0 |  |
| Soil                     |    |       |                   |      |      |                   |      |      |       |                    |         |       |     |  |
| Soil 1                   | 15 | 0.52  | ±                 | 0.03 | 0.53 | ±                 | 0.10 | 29.0 | ±     | 4.6                | 5.9     | ±     | 0.2 |  |
| Soil 2                   | 15 | 1.29  | $\pm$             | 0.10 | 2.18 | $\pm$             | 0.41 | 26.6 | $\pm$ | 5.1                | 4.3     | $\pm$ | 0   |  |
| Soil 3                   | 15 | 1.24  | ±                 | 0.10 | 2.33 | ±                 | 0.34 | 20.9 | ±     | 4.5                | 5.1     | ±     | 0.  |  |
| Soil 4                   | 15 | 1.36  | $\pm$             | 0.11 | 2.64 | $\pm$             | 0.36 | 20.5 | $\pm$ | 4.5                | 3.8     | $\pm$ | 0.  |  |
| Soil 5                   | 15 | 1.65  | $\pm$             | 0.06 | 3.51 | $\pm$             | 0.33 | 44.8 | $\pm$ | 4.9                | 9.4     | $\pm$ | 0.2 |  |
| Soil 6                   | 15 | 1.91  | ±                 | 0.10 | 3.62 | ±                 | 0.38 | 26.1 | ±     | 3.6                | 7.3     | ±     | 0.  |  |
| Soil 7                   | 15 | 1.43  | $\pm$             | 0.07 | 2.63 | $\pm$             | 0.40 | 21.9 | $\pm$ | 2.4                | 49.1    | $\pm$ | 1.  |  |
| Soil 8                   | 15 | 1.42  | $\pm$             | 0.14 | 2.31 | $\pm$             | 0.43 | 21.0 | $\pm$ | 2.7                | 32.7    | $\pm$ | 4.  |  |
| Soil 9                   | 15 | 1.42  | $\pm$             | 0.09 | 2.13 | $\pm$             | 0.29 | 15.3 | $\pm$ | 1.8                | 52.6    | $\pm$ | 3.  |  |
| Soil 10                  | 15 | 1.60  | $\pm$             | 0.08 | 3.26 | $\pm$             | 0.62 | 19.6 | $\pm$ | 1.9                | 49.9    | $\pm$ | 3.4 |  |
| Soil 11                  | 15 | 1.83  | ±                 | 0.08 | 3.26 | ±                 | 0.44 | 20.3 | ±     | 3.2                | 14.9    | ±     | 0.  |  |
| Soil 12                  | 15 | 1.53  | ±                 | 0.06 | 2.97 | ±                 | 0.43 | 32.8 | ±     | 5.2                | 5.6     | ±     | 0.  |  |
| ANOVA                    | P  | value |                   |      |      |                   |      |      |       |                    |         |       |     |  |
| Fertilizer treatment (A) |    | < 0.0 | 001               |      | < 0  | .001              |      | < 0  | .001  |                    | < 0.001 |       |     |  |
| Soil (B)                 |    | < 0.0 | 001               |      | < 0  | .001              |      | < 0  | .001  |                    | < 0.001 |       |     |  |
| AxB                      |    | < 0.0 | 001               |      | < 0  | .001              |      | < 0  | .001  |                    | < 0.    | 001   |     |  |

Mean ± standard error

Number after fertilizer indicates the P rate in mg kg-1 of soil

Mean comparison is not performed since the interaction of both factors is significant. In that case, the effect of one factor depends on the other and an analysis of the effect of main factor cannot be performed

Table 4. Effect of soils on the mineral P fertilizer replacement value (PFRV) expressed in % estimated based on different approaches (DM yield. P uptake. Olsen P) and on the increase in the DTPA extractable Fe after crop for both P fertilizer rates (50 and 100 mg P  $kg^{-1}$  soil)

| Soil type  | n                       | $PFFRV_{DM}$ |      |     |        |       |    |      | V <sub>Puptake</sub> |    |       | PFR  | V <sub>OlsenP</sub> |        | Increase in DTPA extractable Fe |           |        |      |                  |        |      |          |        |    |                  |
|------------|-------------------------|--------------|------|-----|--------|-------|----|------|----------------------|----|-------|------|---------------------|--------|---------------------------------|-----------|--------|------|------------------|--------|------|----------|--------|----|------------------|
|            |                         | 50 m         | g kg | -1  | 100 m  | ng kg | -1 | 50 m | g kg                 | -1 | 100 n | ng k | g <sup>-1</sup>     | 50 r   | ng l                            | $kg^{-1}$ | 100 r  | ng i | kg <sup>-1</sup> | 50 n   | ng k | $g^{-1}$ | 100    | mg | kg <sup>-1</sup> |
| Non-       |                         |              |      |     |        |       |    |      |                      |    |       |      |                     |        |                                 |           |        |      |                  |        |      |          |        |    |                  |
| calcareous | 6                       | 0            | ± 20 | )   | 13     | ±     | 25 | 9.3  | ±                    | 7  | 7.6   | ±    | 3                   | 33     | $\pm$                           | 21        | 30     | ±    | 33               | 10     | ±    | 4.2      | 16     | ±  | 6.5              |
| Calcareous | 6                       | 2.2          | ± 12 | 2   | -41    | ±     | 27 | - 22 | ±                    | 21 | -4.2  | ±    | 14                  | -3.6   | ±                               | 7.8       | 0      | ±    | 3.6              | 0      | ±    | 0.1      | 0.6    | ±  | 0.3              |
|            |                         | ANOV         | /A P | val | ue     |       |    |      |                      |    |       |      |                     |        |                                 |           |        |      |                  |        |      |          |        |    |                  |
|            |                         | NS           |      |     | NS     |       |    | NS   |                      |    | NS    |      |                     | NS     |                                 |           | NS     |      |                  | p < 0. | 01   |          | P<0.0  | )5 |                  |
| Soil pH    |                         |              |      |     |        |       |    |      |                      |    |       |      |                     |        |                                 |           |        |      |                  |        |      |          |        |    |                  |
| < 6.6      | 4                       | 26           | ± 1  | 1   | 40     | ±     | 6  | 16   | ±                    | 6  | 11    | ±    | 3.1                 | 61     | ±                               | 16        | 49     | ±    | 12               | 15     | ±    | 4.8      | 24     | ±  | 6.5              |
| > 7.86     | 8                       | -13          | ± 1: | 5   | -41    | ±     | 23 | -18  | ±                    | 16 | -2.8  | ±    | 10                  | -8.5   | ±                               | 7.5       | -2.1   | ±    | 2                | 0.2    | ±    | 0.2      | 0.5    | ±  | 0.2              |
|            | Kruskall-Wallis P value |              |      |     |        |       |    |      |                      |    |       |      |                     |        |                                 |           |        |      |                  |        |      |          |        |    |                  |
|            |                         | NS           |      |     | 

a

b

 $PFRV_{DM}50 = 6.4 - 3.4 X$ 

Figure 1. Relationship between the P fertilizer replacement value on a dry matter basis at 50 (PFRV<sub>DM50</sub>) (a) and at 100 mg P 255  $kg^{-1}$  (PFRV<sub>DM100</sub>) (b) and the difference between the initial Olsen P in soil and the estimated threshold value (Olsen P – TV).

#### 3.4 Phosphorus Fertilizer Replacement Value on a P Uptake Basis




The P fertilizer replacement value on a P uptake basis (PFRV<sub>P Uptake</sub>) was significantly affected by soil (Table S2). However, when soils were grouped by carbonate content or pH, the effect of soil type was not significant (Table 4). At the lowest P fertilizer rate, the relationship between PFRV<sub>P Uptake</sub> and soil properties was similar to that observed for PFRV<sub>DM</sub>: it was negatively correlated with clay content and Olsen P – TV (r = -0.76 and – 0.77, respectively, P < 0.01 in both cases; an outlier with PFRV<sub>P Uptake</sub> < –130 excluded). The values of PFRV<sub>P Uptake</sub> for the lowest P fertilizer rate were in most cases above 0, with an average of 16 % for Olsen P – TV less than 0 and -18 % for Olsen P – TV higher than 0 (Figure 2). At the highest P fertilizer rate, PFRV<sub>P Uptake</sub> was not related to any soil property, and its values ranged between –50 and 50, with an average of 11 % and –3 % for soils with Olsen P – TV less and higher than 0, respectively. When clay content and Olsen P – TV were considered in a multiple regression, both explained 53 % of the variance in PFRV<sub>P Uptake</sub> at the highest rate (Y = -40.8 - 2.9) (Olsen P – TV) + 0.24 Clay<sup>2</sup>; R<sup>2</sup> = 0.53; P 

 $PFRV_{P \text{ Uptake}} 50 = -0.7 - 2.4 \text{ X}$ 

Figure 2. Relationship between the P fertilizer replacement value on a P uptake basis at 50 mg P kg-1 (PFRV<sub>P Uptake</sub> 50) and the difference between the initial Olsen P in soil and the estimated threshold value (Olsen P - TV).

#### 275 3.5 Phosphorus Fertilizer Replacement Value on an Olsen P Basis

The P fertilizer replacement value on an Olsen P basis (PFRV<sub>P OlsenP</sub>) was also significantly affected by soil (Table S2). When soils were differentiated between calcareous and non-calcareous, differences in this index between both types of soil were not significant at both P fertilizer rates (Table 4). However, when discrimination was done on a pH basis, soils with pH < 6.6 had significantly higher PFRV<sub>P Olsen P</sub> than those with pH > 7.86 at both P fertilizer rates (Table 4). The PFRV<sub>Olsen P</sub> at the lowest P fertilizer rate was negatively correlated with clay content (r = -0.73; P < 0.01), pH (r = -0.80; P < 0.01), and Olsen P – TV (r = -0.83; P < 0.001). Correlations were similar for PFRV<sub>Olsen P</sub> at the highest P fertilizer rate: clay content (r = -0.65; P < 0.05), pH (r = -0.86; P < 0.001), and Olsen P – TV (r = -0.93; P < 0.001). Overall, the highest correlation coefficients were observed for Olsen P – TV, which explained 69 and 87 % of the variance at the lowest and the highest P fertilizer rate, respectively (Figure 3). For Olsen P – TV less than 0, the average PFRV<sub>Olsen P</sub> for the lowest and the highest fertilizer rate was 61 and 49 %, respectively; meanwhile, for Olsen P – TV higher than 0, it was -8.5 and -2.1 %, respectively.

#### 3.6. Increment in DTPA Extractable Fe






The effect of soil on the increment in DTPA extractable Fe with vivianite application relative to the control without fertilizer application was very significant (Table S2). While the increase was negligible in calcareous soils or in soil with pH >7.86, this increase was significant in non-calcareous soils or in soils with pH 

PFRV<sub>Olsen P</sub> 50 = 24.5 – 4.2 X


 $PFRV_{Olsen P} 100 = 22.5 - 3.2 X$ 

Figure 3. Relationship between the P fertilizer replacement value on an Olsen P basis at 50 (PFRV<sub>Olsen P</sub> 50) (a) and 100 mg P kg-1 (PFRV<sub>Olsen P</sub> 100) (b) and the difference between initial Olsen P in soil and the estimated threshold value (Olsen P – TV).

Figure 4. Relationship between the increase of DTPA extractable Fe at 50 (PFRV $_{DM50}$ ) (a) and 100 mg P kg-1 (PFRV $_{DM100}$ ) (b) and the difference between the initial Olsen P in soil and the estimated threshold value (Olsen P - TV)

#### 4. Discussion






Although the effect of the mineral soluble fertilizer on DM yield, P uptake, and Olsen P outperformed that of vivianite, this product achieved a P fertilizer replacement value on a DM basis (PFRV<sub>DM</sub>) of around 50 % in some soils. This means that, in these soils, vivianite can replace half of the soluble mineral fertilizer, implying a promising result for its use as a P fertilizer. The assessment of the efficiency of vivianite relative to mineral fertilizer should be done based on the P fertilizer replacement value (PFRV) since soil properties differentially affect the fate of both soluble mineral fertilizer and vivianite. The PFRV estimated with the three approaches (DM, P uptake, and Olsen P) ranged widely among soils. This reveals a different effect of soil properties on the dynamics and, consequently, on the efficiency of both types of fertilizers. In particular, the best results of vivianite on PFRV<sub>DM</sub> were obtained in the four soils with acidic pH (< 6.6) and in two calcareous soils (Soil 3 and 4) (Table S2). The highest increase of DTPA Fe with vivianite relative to control and mineral soluble fertilizer was observed in three acidic soils. This reveals that conditions prone to vivianite dissolution, i.e., acidic pH in soils, determine its efficiency as a P and Fe fertilizer.

Overall, results were lower when the PFRV was estimated on a P uptake basis, and only values above 20 % were found in two soils at the lowest P fertilizer rate (Figure 2). Lower PFRV values on a P uptake basis can be expected since, with the P rates applied, particularly the highest P rate, a luxury consumption can be promoted, i.e., P accumulation in plants exceeding the minimal for maximum DM yield (Penn et al., 2023). On the other hand, negative values are expected when the P supply capacity of soils is high enough to cover crop needs, and this explains that, frequently, PFRV values were negative when Olsen P – TV was higher than 0 (soil P test above the threshold value), that is, when soil P is expected to cover crop needs. An increase in Olsen P above the P threshold value did not result in a further increase in crop yield (Johnson, 2001; 2005; Tandy et al., 2021). This could partly explain why soils with an already high P status in the current study did not lead to an increased PFRV.

An analysis of the PFRV on an Olsen P basis allows one to think that results could even be more positive with the application of vivianite as P fertilizer since average values were 49 and 61 % at the highest and lowest P rate, respectively, when the soil Olsen P was below the threshold value which corresponded with soils with pH lower than 6.6 (Table 4). This is much higher than PFRV on a DM or P uptake basis and reveals that the long-term effect of vivianite, beyond the studied crop cycle, could be very interesting in acidic soils with P levels below the threshold value. Thus, one short-term growing cycle probably does not fully reflect the potential of vivianite as a P fertilizer, which can have a relevant residual effect according to the effect on the soil P test in soils with low P status.

It can be supposed that the PFRV of vivianite was determined by its solubility, with a crop response above the threshold values not being expected. However, the values of the PFRV on an Olsen P basis reveal that there is limited solubilization when the soil Olsen P value is above the threshold value (PFRV<sub>Olsen P</sub> around 0). Theoretically, for soluble fertilizers such as superphosphate, solubilization is not necessarily limited in soils with Olsen P above the threshold value. These results with Olsen P after harvesting agree with the observed increase in DTPA extractable Fe. This increased DTPA extractable Fe comes from the dissolution of vivianite (de Santiago & Delgado, 2010). This increase was negligible when the soil Olsen P was above

the threshold value. Thus, it seems that fertilizer dissolution determined the response of crops to applied vivianite, and this dissolution was expected to be increased at acidic pH (pH < 6.6), which corresponded to soils with Olsen P values below the threshold value for fertilizer response. In fact, the highest PFRV on Olsen P basis (around 100 %) was found in the more acidic soil. Metz et al. (2023) found that the dissolution rate of vivianite under anoxic conditions increased strongly with a decreasing pH, and at pH 5, all solid materials of vivianite were found to have completely dissolved. This observation was similar when a vivianite dissolution experiment was conducted under oxic conditions (personal communication with Rouven Metz). This could invariably mean that an acidic pH favours the dissolution of vivianite, leading to the release of P from vivianite, thereby making P available in the soil solution where plants can take up P. This situation seems to be different in alkaline calcareous soils (the other eight soils with pH > 7.86) because of a lower rate of dissolution (Metz et al., 2023).

However, pH should not be the only factor affecting P recovery from vivianite. In fact, soil 4 was calcareous and showed a PFRV $_{DM}$  of 34 % at the lowest fertilizer rate. This was also the only calcareous soil in which vivianite at both rates increased DM yield relative to control (Table S1). In addition, PFRV on a DM and P uptake basis was related to Olsen P - TV but not to pH. Thus, it seems that soil pH may have a crucial role in the dissolution of vivianite, as mentioned above, but there are other factors contributing to its use as fertilizer by crops, in particular, the available P status of soil (reflected in the Olsen P - TV values). This difference between Olsen P and threshold value explained more variance in the PFRV on DM and P uptake basis than pH. In any case, it is not easy to separate the effect of soil pH from that of low P availability since pH and Olsen P - TV were positively correlated.

In P-limiting soils, mechanisms to obtain adequate P for growth are triggered (Raghothama & Karthikeyan, 2005; Balemi & Negisho, 2012). This involves the modification of the plant root system (Lynch, 2011; López-Arredondo et al., 2014) and the increased exudation of organic acids (Neumann & Römheld, 1999; Dechassa & Schenk, 2004), which promotes the mobilization of poorly soluble P from soil (Kpomblekou-A & Tabatabai, 2003; Johnson & Loeppert, 2006). According to Schütze et al. (2020), organic ligands released by roots, such as citrate, enhance the dissolution of vivianite. Talboys et al. (2016) observed an increased organic acid concentration in the rhizosphere when struvite, a poorly soluble P compound, was supplied as a P fertilizer instead of soluble fertilizers. Thus, when a poorly soluble fertilizer such as vivianite is applied, an increased expression of P mobilizing mechanisms in P-poor soils can be expected. This contributes to explaining the increase in PFRV with decreased Olsen P – TV values.

The role of soil microorganisms in the rhizosphere in promoting the dissolution and use of P from vivianite by crops cannot be ruled out. They play an important role in the solubilization and mobilization of P ((Richardson, 2007; García-López et al., 2018; 2021), thus increasing the bioavailability of P (Deubel & Merbach, 2005), especially in P-poor soils. In P-deficient soils, microbial communities are often dominated by phosphate-solubilizing bacteria and fungi, capable of producing organic acids and enzymes that solubilize P, thus making it available for plant uptake (Weigh et al., 2023). This is not always the case under P-abundant conditions (R. Sun et al., 2022; Yadav & Yadav, 2024). Thus, the particular structure of microbial communities in P-deficient soils can contribute to better dissolution of vivianite and, consequently, to its efficiency as a P fertilizer.

The negative correlation observed between replacement values and clay content in some cases can be determined by the correlation between this soil property and the Olsen P – TV values. In addition, the soils with pH above neutrality had the highest clay content. Furthermore, clay is a soil property usually positively correlated with P buffer capacity and P adsorption, thus affecting P dynamics and availability to plants (Recena et al., 2015; 2016).

In the current study, the DTPA extractable Fe supports the increased dissolution of vivianite under acidic conditions, which were also the soils with the lowest P availability to plants. Vivianite has a considerable content of Fe (Eynard et al., 1992), and the release of Fe is expected following its dissolution. However, at alkaline pH, there is a preferential release of P over Fe leading to the structural oxidation of Fe and the subsequent formation of a Fe(III)-bearing phosphate phase (Thinnappan et al., 2008). In fact, the efficiency of synthetic vivianite as a source of Fe for plants in calcareous soils (Rombolà et al., 2003; Díaz et al., 2009), has been ascribed to the formation of poorly crystalline oxides such as ferrihydrite and lepidocrocite (Eynard et al., 1992; Roldán et al., 2002). These oxides have a high specific surface and high P adsorption capacity. Thus, reaction products of vivianite dissolution in alkaline soils can contribute to a decreased PFRV in these soils.

The results obtained are promising with a view of using vivianite from water purification treatments as P fertilizer. Its efficiency as fertilizer was in line with other fertilizers obtained from recycling (Hernández-Mora et al., 2024; Frick et al., 2025). However, this efficiency varies greatly depending on soil properties and reveals the need of tailoring its use to specific soil conditions. Although further field research is necessary for more solid recommendations, based on the present results, it seems to be a recommendable fertilizer in acidic P-poor soils. Even in non-acidic P-poor soils, it can have positive effects as a P fertilizer. An advantage is that its dissolution seems to be enhanced by plants and microbial P-mobilizing mechanisms that are triggered in P-poor soils when poorly soluble fertilizers are applied. This means that its dissolution is faster when plants are present, thus reducing environmental risks when soils have a low P adsorption capacity. Since it is a poorly soluble product, vivianite should be incorporated into the soil close to the areas of maximum root development to enhance its solubilization and use by plants, as previously done for recommendation as Fe fertilizer (Rosado et al., 2002; Díaz et al., 2009). However, in the EU, for its practical use and recommendation, a normative change is necessary to avoid restrictions on the use of products with high Fe content. From an economic point of view, P recovery from wastewater through vivianite precipitation is gaining interest since its separation is easier than other byproducts, such as struvite, through its magnetic properties, and this can contribute to a more competitive price as a fertilizer.

#### 5. Conclusions







Overall, vivianite was not as efficient as P fertilizer as a soluble mineral fertilizer. The application of vivianite as a P fertilizer was more effective in acidic soils with soil P tests below the threshold value for fertilizer response. The effect of vivianite on dry matter yield could be equivalent, on average, to 40 % of the same amount applied as mineral soluble fertilizer in these soils. The effect on Olsen P in soil could be equivalent, on average, to 61 % of the same amount applied as soluble mineral fertilizer. This is explained not only by the increased solubility of this fertilizer under acidic conditions but also by a low P availability to plants, which can trigger plant and microbial mobilization mechanisms, leading to increased efficiency of

vivianite as a P fertilizer. These results are encouraging for promoting the use of vivianite from wastewater treatment as a P fertilizer, the application of which should be adapted to the soil properties, and is especially recommended for acidic soils low in P.

# **Data Availability Statement**

The data that support the findings of this study are available from the corresponding author upon reasonable request.

#### 450 Author Contributions

Conceptualization, T.A., R.R. and A.D.; data curation, T.A.; formal analysis, T.A., J.M.Q. and A.D; investigation, T.A., R.R., J.M.Q., A.M.G.-L. and A.D.; methodology, T.A., R.R. and A.M.G.-L.; supervision, A.D. and M.C.d.C.; writing—original draft, T.A., M.C.d.C. and A.D.; writing—review and editing, T.A., A.M.G.L., M.C.d.C. and A.D. All authors have read and agreed to the published version of the manuscript.

#### **Conflicts of Interest**



The authors declare no conflicts of interest.

# Acknowledgements

The authors would like to appreciate Wetsus (a non-academic partner—NAPO) of the P-TRAP project for providing the vivianite from water purification used in this study. Our sincere gratitude also goes to Vidal Barrón for assisting with the characterization of the fertilizer product used in the study. We would also like to thank Prof. Dr Erik Smolders for the opportunity to carry out some of the laboratory analysis at his laboratory during a 3-month secondment at KU Leuven in Belgium.

#### **Funding**

This research was funded by the European Union's Horizon 2020 Research and Innovation Programme under the Marie Sklodowska-Curie grant agreement No 813438 and is a part of the P-TRAP (Diffuse phosphorus input to surface waters—new concepts in removal, recycling, and management) Project.

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

# 755 Figure captions

- Figure 1. Relationship between the P fertilizer replacement value on a dry matter basis at 50 (PFRV<sub>DM50</sub>) (a) and 100 mg P kg<sup>-1</sup> (PFRV<sub>DM100</sub>) (b) and the difference between the initial Olsen P in soil and the estimated threshold value (Olsen P TV).
- Figure 2. Relationship between the P fertilizer replacement value on a P uptake basis at 50 mg P kg<sup>-1</sup> (PFRV<sub>P Uptake50</sub>) and the difference between the initial Olsen P in soil and the estimated threshold value (Olsen P TV).
  - Figure 3. Relationship between the P fertilizer replacement value on an Olsen P basis at 50 (PFRV<sub>Olsen P</sub> 50) (a) and 100 mg P  $kg^{-1}$  (PFRV<sub>Olsen P</sub> 100) (b) and the difference between initial Olsen P in soil and the estimated threshold value (Olsen P TV).
- Figure 4. Relationship between the increase of DTPA extractable Fe at 50 (PFRV<sub>DM50</sub>) (a) and 100 mg P kg<sup>-1</sup> (PFRV<sub>DM100</sub>) (b) and the difference between the initial Olsen P in soil and the estimated threshold value (Olsen P TV).