# Peer review of "Circular Economy Approach in Phosphorus Fertilization Based on Vivianite must be Tailored to Soil Properties"

_EGUsphere, 2025_

## Author Response (AR1)

**SOIL Journal Report Response - Response to Topic Editors and Reviewers**

**General comment by Topic Editor:**

This study aimed to evaluate the effect of soil properties on the efficiency of vivianite recovered from water purification as a P fertilizer and to provide practical recommendations for its effective use. Phosphorus is an essential nutrient for plant growth, but its sources are depleting, and excessive use can cause environmental pollution. The study aims to investigate how vivianite can be utilized in agricultural practices, focusing on its effectiveness across different soil types. The research examines several soil properties, including pH, texture, and organic content, and their influence on the availability of phosphorus from vivianite. The results indicate that vivianite can improve crop yield and phosphorus uptake in soils with varying nutrient levels. This study provides valuable insights into the potential of vivianite as a sustainable phosphorus fertilizer to support sustainable agriculture and the circular economy by recycling phosphorus. I recommend acceptance of this paper for publication with minor revisions.

**General Response:**

**Dear handling editor,**

Many thanks for your kind and insightful comments. We appreciate the opportunity to address the excellent suggestions and calls for clarification in a revised version of our manuscript. We have taken into consideration all comments by the Anonymous Referee, and especially those pointed out by you concerning the improved introduction and discussion sections. We hope that you can now consider our revised manuscript for publication in SOIL.

Yours sincerely,

**Comment**

**Introduction Section:**

Literature Review Insufficiency: the literature mentioned in the introduction, although extensive, does not adequately discuss the advantages and challenges of vivianite as a phosphorus fertilizer, especially when compared to conventional phosphorus fertilizers (e.g., ammonium phosphate) and other phosphorus sources (e.g., calcium oxalate). It is recommended to increase the literature discussion especially on the application of vivianite in agriculture and its potential impact on the environment.

Vague motivation of the study: the motivation section of the study is vague and does not clearly articulate why vivianite was chosen as a material, especially its place in the available fertilizer

options. It is recommended that the innovation of the study be further articulated to highlight the environmental and economic advantages of vivianite.

**Response:**

According to the suggestion, we have improved and changed this section, implementing recent bibliography and justifying our study. We have put special emphasis, following the recommendation, on the arguments to select vivianite and the evidence of its potential as a source of nutrients for plants. Here, it is relevant to mention the relevance of wastewater as a nutrient-rich flow that is necessary to include in circular economy strategies for increased sustainable use of nutrients at a societal scale. We have improved the literature review on the use of vivianite in agriculture and also provided some insights on its properties to fully understand its function as a fertilizer.

**Comment:**

**Discussion Section:**

It is recommended that the authors include in the discussion section specific recommendations for practical applications, especially best practices for the application of vivianite. For example, how to adjust the amount and frequency of application to suit different soil types, and how to incorporate other soil amendment techniques (e.g., organic fertilizers, lime application, etc.) to improve the efficacy of phosphate fertilizers. Discussions could also be held on how to promote the use of vivianite in agricultural policies, taking into account its environmental and economic advantages.

**Response:**

We agree with the comment, and a paragraph in the discussion dealing with practical recommendations has been included. This discusses its efficiency, the need to adapt to soil types, application methods, the economical implications, the environmental benefits, and the regulatory framework for promoting its use.

**Comment:**

It is recommended to include in the discussion the potential impacts of vivianite application on the structure of soil microbial communities and how it affects the availability of phosphorus in the soil. For example, is the phosphorus in vivianite better converted into plant-available forms through microbial metabolism? This information will round out the discussion of the article.

**Response:**

We fully agree with this interesting aspect, and the relevant role that microbial activity can have on vivianite dissolution and efficiency as fertilizer as now be considered in the discussion.

**Comment:**

Figure captions Section:

Figure 2.: Recurring issues.

"kg-1" should be modified to "kg-1"

**Response:**

This mistake has been corrected.

**Comment:**

Reminder: The article has a high repetition rate, please reduce the repetition rate appropriately, especially in the Materials and Methods section.

**Response:**

We have tried to reduce repetition rate, but it is difficult with usual chemical methods in soil. We hope this can be taken into account.

**Anonymous Referee #1, 01 Jul 2025**

**General Comment:**

The manuscript presents a pot experiment evaluating the efficacy of vivianite mostly obtained from wastewater treatment plants as a phosphorus (P) fertilizer. The study compares vivianite to a conventional soluble P fertilizer (superphosphate) at two application rates across twelve different soils with a wide range of properties. The authors use wheat as a test crop and measure dry matter (DM) yield, P uptake, and post-harvest soil P (Olsen P) and Fe (DTPA Fe) levels. The central finding is that vivianite's effectiveness is highly dependent on soil properties, performing best in acidic soils that are also deficient in plant-available P. The study introduces and uses the P fertilizer replacement value (PFRV) to quantify vivianite's equivalence to superphosphate. A key conclusion is that vivianite is more effective in acidic soils having P below threshold value. Overall, manuscript is well structured and written. My biggest comments are for introduction and discussion sections.

**Response:**

Thank you very much for your kind, positive and insightful comment on our article. We are also pleased to know that you found our study of great importance.

**Comment:**

The Introduction provides a thorough explanation of soil properties relevant to phosphorus dynamics; however, it offers limited background on vivianite. It would strengthen the manuscript to include more context on the origins of vivianite, specifically, whether wastewater treatment plants are the only viable source. Discussing the major sources of vivianite and highlighting the significance of wastewater streams as a sustainable and potentially abundant supply would provide valuable context. Furthermore, a brief discussion on the economic feasibility of recovering vivianite and its potential advantages over other recycled P sources would be useful. Including its chemical composition, especially the phosphorus content by percentage, would also help readers understand its agronomic value.

**Response:**

We fully agree with the suggestion and the introduction has been changed and improved with your comments as well as the editor's comments, adding an updated bibliography. We have put attention on the justification on the use of vivianite, and we have to remark that its potential practical benefits are also discussed in the discussions section

**Comment:**

The Discussion section is comprehensive, and the authors provide a detailed interpretation of how soil properties, particularly pH and Olsen P status, influence the effectiveness of vivianite as a phosphorus fertilizer. However, the section at times reads like an extended summary of the Introduction and Results sections rather than a focused synthesis of key findings. Several parts reiterate previously stated information, which affects the clarity and conciseness of the discussion. Condensing the text to avoid repetition and emphasizing how the study's findings advance current knowledge would significantly improve the section. Reorganizing the content into thematically coherent and concise paragraphs would enhance readability and help readers more easily grasp the significance of the findings.

**Response:**

We fully agree with the suggestion. Introductory material has been removed from the discussion and references are discusses as support to our conclusions. We have put the focus on the most relevant results and in their explanation. We have tried to improve readability with better structured information and more concise paragraphs

**Comment:**

Although the abstract states that the study aims to provide "practical recommendations for vivianite's effective use," these recommendations are not clearly presented in the manuscript. The insights about soil pH, Olsen P status, and P fertilizer replacement value are valuable, but they are scattered throughout the results and discussion. To better fulfill the stated objective, the authors should consider summarizing these findings into a concise, standalone section—either as a paragraph in the discussion or a separate recommendations section—outlining when, where, and how vivianite should be applied, possibly with threshold guidelines or a decision matrix. This would greatly enhance the practical relevance and clarity of the manuscript.

**Response:**

We fully agree with this recommendation; a paragraph has been included in the discussion; see answer to editor.

**Anonymous Referee #2, 18 Aug 2025**

**General Comment:**

This study evaluates whether vivianite recovered from wastewater can substitute mineral superphosphate across 12 soils with varying properties by measuring dry matter yield, phosphorus uptake and Olsen P; the topic aligns with journals focusing on nutrient cycling and circular economy. The use of vivianite as a phosphorus source has been reported, but previous studies rarely compared its performance across multiple soil types; the present investigation leverages 12 soils, two fertilizer rates and a replacement-value approach. Comparative multi-soil testing adds value. The authors used a randomized complete block design with 12 contrasting soils and three replicates per treatment. Each pot contained one wheat plant receiving one of two P rates (50 and 100 mg kg-1) of either vivianite or superphosphate plus an unfertilized control. The experiment was carried out under controlled conditions. The Ms. is generally clear and well structured; sentences are coherent and vocabulary appropriate. Occasional minor grammatical slips and typographical errors could be corrected, but they do not impede comprehension.

**Response**

We greatly appreciate these positive comments

My main concerns are as follows:

**Comment:**

Extrapolating short pot experiments to field recommendations seems premature; a fuller discussion of limitations and negative responses would strengthen the Ms.

**Response:**

We fully agree and have taken this into account and remarked the need of field experiments for more solid recommendations

**Comment:**

The authors should clarify how negative replacement values should be interpreted, and temper field recommendations. A deeper discussion of soil microbial factors and a clearer statement of what constitutes an agronomically acceptable replacement value would strengthen the Ms.

**Response:**

We fully agree with this comment. The negative values, in particular with PFRV on P uptake basis has been explained and interpreted. Microbial factors have been also considered as explained to editor comments. Finally, we have supported agronomically acceptable effect on the grounds with previous published literature with other alternative fertilizers.

**Comment:**

The Results section begins with the sentence: 'There was wide variation in the properties of the set of soils used in this experiment (Table 1), especially clay content and calcium carbonate equivalent (CCE), which are relevant properties affecting P dynamics in soils.' In the Conclusions, the Authors refer to the acidic condition of the investigated soils (and thus mention calcium carbonate content) but make no reference to soil texture. If no relationship was found with texture, this should also be explicitly stated.

**Response:**

The relationship of clay content with PFRV has been described and discussed. In the discussion section, this relationship is also discussed and explained.